# Longitudinal trajectories, correlations and mortality associations of nine biological ages across 20-years follow-up

Xia Li[1], Alexander Ploner[1], Yunzhang Wang[1], Patrik KE Magnusson[1], Chandra Reynolds[2], Deborah Finkel[3,4], Nancy L Pedersen[1], Juulia Jylhävä[1], Sara Hägg[1]*

[1]Department of Medical Epidemiology and Biostatistics, Karolinska Institutet, Stockholm, Sweden; [2]Department of Psychology, University of California, Riverside, Riverside, United States; [3]Department of Psychology, Indiana University Southeast, New Albany, United States; [4]Institute for Gerontology, Jönköping University, Jönköping, Sweden

**Abstract** Biological age measurements (BAs) assess aging-related physiological change and predict health risks among individuals of the same chronological age (CA). Multiple BAs have been proposed and are well studied individually but not jointly. We included 845 individuals and 3973 repeated measurements from a Swedish population-based cohort and examined longitudinal trajectories, correlations, and mortality associations of nine BAs across 20 years follow-up. We found the longitudinal growth of functional BAs accelerated around age 70; average levels of BA curves differed by sex across the age span (50–90 years). All BAs were correlated to varying degrees; correlations were mostly explained by CA. Individually, all BAs except for telomere length were associated with mortality risk independently of CA. The largest effects were seen for methylation age estimators (GrimAge) and the frailty index (FI). In joint models, two methylation age estimators (Horvath and GrimAge) and FI remained predictive, suggesting they are complementary in predicting mortality.

*For correspondence:
Sara.Hagg@ki.se

Competing interests: The authors declare that no competing interests exist.

## Introduction

Biological aging is a process characterized by progressive deteriorations occurring simultaneously at multiple levels, that is molecular, cellular, tissue- and organ-specific, in an organism (*Hayflick, 2002*). Due to its complex nature, quantifying this process has long been a challenge. Chronological age (CA) records the passage of time and serves as a convenient metric of aging; however, people of the same CA manifest diverse aging-related phenotypes. Hence, biological age measurements (BAs), attempt to capture physiological changes along the aging course. Therefore, they can be used to assess health risks in individuals of the same age. Multiple BAs have been proposed, including omics-based clocks, clinical biomarkers, and assessments of functioning (*Jylhävä et al., 2017*; *Xia et al., 2017*). One of the potential BAs, telomere length, is the length of a nucleotide sequence at the chromosomal ends, and represents the capacity for cell division (*Mather et al., 2011*). Epigenetic clocks comprise aging-related DNA methylation information from various genomic loci (i.e., CpG dinucleotides) (*Hannum et al., 2013*; *Horvath, 2013*). Another composite score derived from clinical measurements and blood biomarkers can be viewed as a physiological age measure (*Levine, 2013*). Furthermore, organismal functioning status could reflect the biological aging process from various aspects, such as cognitive performance, physical functioning, and overall frailty (*Reynolds et al., 2005*; *Finkel et al., 2019*; *Searle et al., 2008*).

**eLife digest** Everyone ages, but how aging affects health varies from person to person. This means that how old someone seems or feels does not always match the number of years they have been alive; in other words, someone's "biological age" can often differ from their "chronological age".

Scientists are now looking at the physiological changes related to aging to better predict who is at the greatest risk of age-related health problems. Several measurements of biological age have been put forward to capture information about various age-related changes. For example, some measurements look at changes to DNA molecules, while others measure signs of frailty, or deterioration in cognitive or physical abilities. However, to date, most studies into measures of biological age have looked at them individually and less is known about how these physiological changes interact, which is likely to be important.

Now, Li et al. have looked at data on nine different measures of biological age in a group of 845 Swedish adults, aged between 50 and 90, that was collected several times over a follow-up period of about 20 years. The dataset also gave details of the individuals' birth year, sex, height, weight, smoking status, and education. The year of death was also collected from national registers for all individual in the group who had since died.

Li et al. found that all nine biological age measures could be used to explain the risk of individuals in the group dying during the follow-up period. In other words, when comparing individuals with the same chronological age in the group under study, the person with a higher biological age measure was more likely to die earlier. The analysis also revealed that biological aging appears to accelerate as individuals approach 70 years old, and that there are noticeable differences in the aging process between men and women.

Lastly, when combining all nine biological age measures, some of them worked better than others. Measurements of methylation groups added to DNA (known as DNA methylation age) and frailty (the frailty index) led to improved predictions for an individual's risk of death. Ultimately, if future studies confirm these results for measures from single individuals, DNA methylation and the frailty index may be used to help identify people who may benefit the most from interventions to prevent age-related health conditions.

Previous studies found low to moderate correlations among different types of BAs, for example telomeres, epigenetic clocks, clinical biomarkers, and frailty (*Belsky et al., 2018*; *Kim et al., 2017*; *Vetter et al., 2019*; *Zhang et al., 2018*). However, few studies comparing BAs have taken mortality risk into consideration, and none of them have included more than three types of BAs (*Kim et al., 2017*; *Zhang et al., 2018*; *Murabito et al., 2018*). Thus, to grasp more of the complex aging process, different types of BAs should be studied jointly in the same cohort. In addition, to capture changes over time, as in aging, it is imperative to study the same individuals longitudinally, yet there is a lack of studies comparing longitudinal profiles of multiple BAs in the same population.

Therefore, the present study aims to add to this knowledge gap in the field by examining longitudinal trajectories, correlations and associations to all-cause mortality using nine BAs in the Swedish Adoption/Twin Study of Aging (SATSA) (*Finkel and Pedersen, 2004*). Here, telomere length, four types of DNA methylation age estimators (DNAmAges), physiological age, cognitive function, functional aging index (FAI), and frailty index (FI) have been measured repeatedly across 20 years follow up.

## Results

In the present study, 845 individuals in SATSA (*Finkel and Pedersen, 2004*), with at least one BA assessment at one In-Person Testing (IPT) occasion, were included. In total, there were nine IPTs conducted between 1986 and 2014, that is IPT1-3 and IPT5-10 (IPT4 was a telephone interview). At each IPT, a maximum of nine BAs were collected, resulting in 3973 BA measurements overall. A number of 288 persons had at least one IPT with complete BA information on all nine measurements (referred to as 'complete measurement' hereafter); in total, 612 complete measurements were included. (*Table 1* and *Supplementary file 1A*).

SATSA was initially designed to study individual differences in aging within twin pairs reared apart. The present study, however, did not focus on twin-oriented aspects, such as quantifying the magnitude of familial effect in exploring the BA-mortality associations. Instead, we made statistical inferences for the general population, in which relatedness within twin pairs was accounted for by introducing random effects in the mixed models and estimating robust errors in Cox regressions.

## Overview of the development of BAs and BA residuals

Nine BAs were included in the present study based on data availability in SATSA. Leukocyte *telomere length* was measured by quantitative polymerase chain reaction (qPCR) and presented as a relative telomere length (*Berglund et al., 2016*). Genome-wide methylation levels in whole blood were assessed by Infinium HumanMethylation 450K BeadChip, and four types of *DNAmAge* (Horvath, Hannum, PhenoAge, and GrimAge) were then calculated using established algorithms based on the methylation levels of 353, 71, 513, and 1030 CpG sites, respectively (*Hannum et al., 2013*; *Horvath, 2013*; *Levine et al., 2018*; *Lu et al., 2019*). *Physiological age* was derived from a list of age-correlated biomarkers (Pearson correlation coefficient with CA > 0.10) measured by blood tests and physical examinations (see Materials and methods and *Supplementary file 1B*). Biomarkers were transformed into a calibrated BA value using principal component analysis and the *Klemera and Doubal (2006)* methods. *Cognitive function* measured performance in verbal (crystallized) ability, spatial (fluid) ability, memory, and perceptual speed using a cognitive testing battery. The four specific domains were combined into an overall score by principal component analysis (*Reynolds et al., 2005*). *FAI* incorporated four functional aspects from self-reported information and physical examinations, that is sensory (vision and hearing), pulmonary, strength (grip strength), and movement/balance (gait speed). The four indicators were standardized separately and summed to create a composite score (*Finkel et al., 2019*). Lastly, a set of 42 self-reported deficits, covering a range of health domains, were taken into account in the development of the *FI* (*Jiang et al., 2017*). (*Table 2* and *Supplementary file 1B– C*).

For each BA, we calculated a corresponding BA residual by regressing out CA and sex effects as well as individual- and twin-pair related components from the BA using a linear mixed model (see Materials and methods and *Figure 1*). The BA residuals in the present study are commonly referred to as age acceleration in the DNAmAge-related literature.

## Baseline characteristics of BAs

Of the 845 study participants, 59.5% were women, 42.2% attained above primary education, and 25.0% were current or former smokers; the average body mass index (BMI) was 25.7 kg/m², and mean chronological age was 63.6 years at baseline. On average, the levels of BAs were 0.73 for telomere length (T/S ratio), 60.4 years for Horvath DNAmAge, 65.2 years for Hannum DNAmAge, 63.8 years for DNAmPhenoAge, 69.4 years for DNAmGrimAge, 64.7 years for physiological age, 51.5 for

**Table 1.** Number of individuals and measurements with information on BAs.

|  | Individuals | Measurements | Included IPTs |
|---|---|---|---|
| Telomere length | 636 | 1599 | 3, 5, 6, 8, 9 |
| DNAmAge (four types) | 387 | 1028 | 3, 5, 6, 8, 9 |
| Physiological age | 802 | 3175 | 1, 2, 3, 5, 6, 7, 8, 9, 10 |
| Cognitive function | 829 | 3045 | 1, 2, 3, 5, 6, 7, 8, 9 |
| FAI | 739 | 2922 | 2, 3, 5, 6, 7, 8, 9, 10 |
| FI | 756 | 3162 | 2, 3, 5, 6, 7, 8, 9, 10 |
| At least one BA | 845 | 3973 | 1, 2, 3, 5, 6, 7, 8, 9, 10 |
| Complete nine BAs | 288 | 612 | 3, 5, 6, 8, 9 |

One measurement refers to one in-person testing occasion with at least one BA assessed in a given individual.

DNAmAges include four different types: Horvath, Hannum, PhenoAge, GrimAge.

BA, biological age; IPT, in-person testing; DNAmAge, DNA methylation age estimator; FAI, functional aging index; FI, frailty index.

**Table 2.** The construction of BAs.

| | Component elements | Measurement | Statistical methods |
|---|---|---|---|
| Telomere length (*Berglund et al., 2016*) | Leukocyte telomere length | qPCR | Ratio of measured telomere length to a reference length |
| DNAmAges: | Leukocyte DNA Methylation levels of: | Infinium HumanMethylation 450K BeadChip | Established clock algorithms, developed by: |
| Horvath (*Horvath, 2013*) | 353 age-associated CpGs | | Elastic net regression (regressing CA on CpGs) |
| Hannum (*Hannum et al., 2013*) | 71 age-associated CpGs | | Elastic net regression (regressing CA on CpGs) |
| PhenoAge (*Levine et al., 2018*) | 513 PhenoAge-associated CpGs | | Step 1: Penalized proportional hazard regression (regressing time-to-death on clinical markers and CA to create PhenoAge); Step 2: Elastic net regression (regressing PhenoAge on CpGs) |
| GrimAge (*Lu et al., 2019*) | Smoking pack-years- and seven plasma proteins-associated CpGs (1030 unique sites) | | Step 1: Elastic net regression (regressing biomarkers on CpGs to develop DNAm-based biomarkers); Step 2: Elastic net Cox regression (regressing time-to-death on DNAm-based biomarkers and CA) |
| Physiological age | Age-related biomarkers, including: **Blood biomarkers**: hemoglobin, glucose, cholesterol, Apolipoprotein B, triglyceride; **Clinical markers**: BMI, waist hip rate, weight, waist circumference, hip circumference, systolic BP, diastolic BP | Blood test and physical examination | Principal component analysis and the Klemera and Doubal methods (*Klemera and Doubal, 2006*) |
| Cognitive function (*Reynolds et al., 2005*) | Verbal (crystallized) ability, spatial (fluid) ability, memory, and perceptual speed | In-person cognitive testing | Principal component analysis |
| FAI (*Finkel et al., 2019*) | Sensory (vision and hearing), pulmonary, strength (grip strength), and movement/balance (gait speed) | Self-reported questionnaire and physical examination | Sum of standardized scores |
| FI (*Jiang et al., 2017*) | 42 health deficits | Self-reported questionnaire | Ratio of the number of deficit presented to the total number of deficit considered |

BA, biological age; IPT, in-person testing; DNAmAge, DNA methylation age estimator; FAI, functional aging index; FI, frailty index; BMI, body mass index; qPCR, quantitative polymerase chain reaction; CpG, cytosine nucleotide being followed by a guanine nucleotide; BP, blood pressure.

cognitive function, 48.3 for FAI, and 0.10 for FI. The baseline characteristics in individuals with complete measurements were similar to those in all individuals. (*Table 3* and *Supplementary file 1D*).

## Longitudinal trajectories of BAs

On average, BAs were assessed between 2.5 times (telomere length) to 4.2 times (FI) in each participant with information on the respective BAs. (*Table 3*) Longitudinal changes in BAs were modeled as functions of CA (as a natural spline with three degrees of freedom) and sex, with random effects introduced at the individual and twin-pair levels (mixed models, see Materials and methods). Both individual-level BAs and population BA means over CA in men and women are presented in *Figure 1*. We found that with increasing age, average telomere length and cognitive function declined, while all types of DNAmAges, physiological age, FAI, and FI increased. The profiles for the three functional BAs (cognitive function, FAI, and FI) show clear curvature, indicating an accelerated rate of change around the age of 70, whereas the other BAs exhibit minimal or no curvature, corresponding to a constant change (linear growth) over the age span. In addition, we observed small to moderate sex differences in the mean levels of BAs. Women exhibited longer telomere length (p=0.001) and lower DNAmAge (p=0.013 for Horvath, p=0.001 for Hannum, and p<0.001 for GrimAge) compared to men, but also worse functioning (p<0.001 for FAI and p=0.001 for FI). (*Figure 1*).

To examine the possible effect modification, we next introduced an interaction term between CA and sex in the mixed models. We found no evidence that the shape of the BA curves differed between men and women, except for in physiological age (p<0.001 for sex interaction). However,

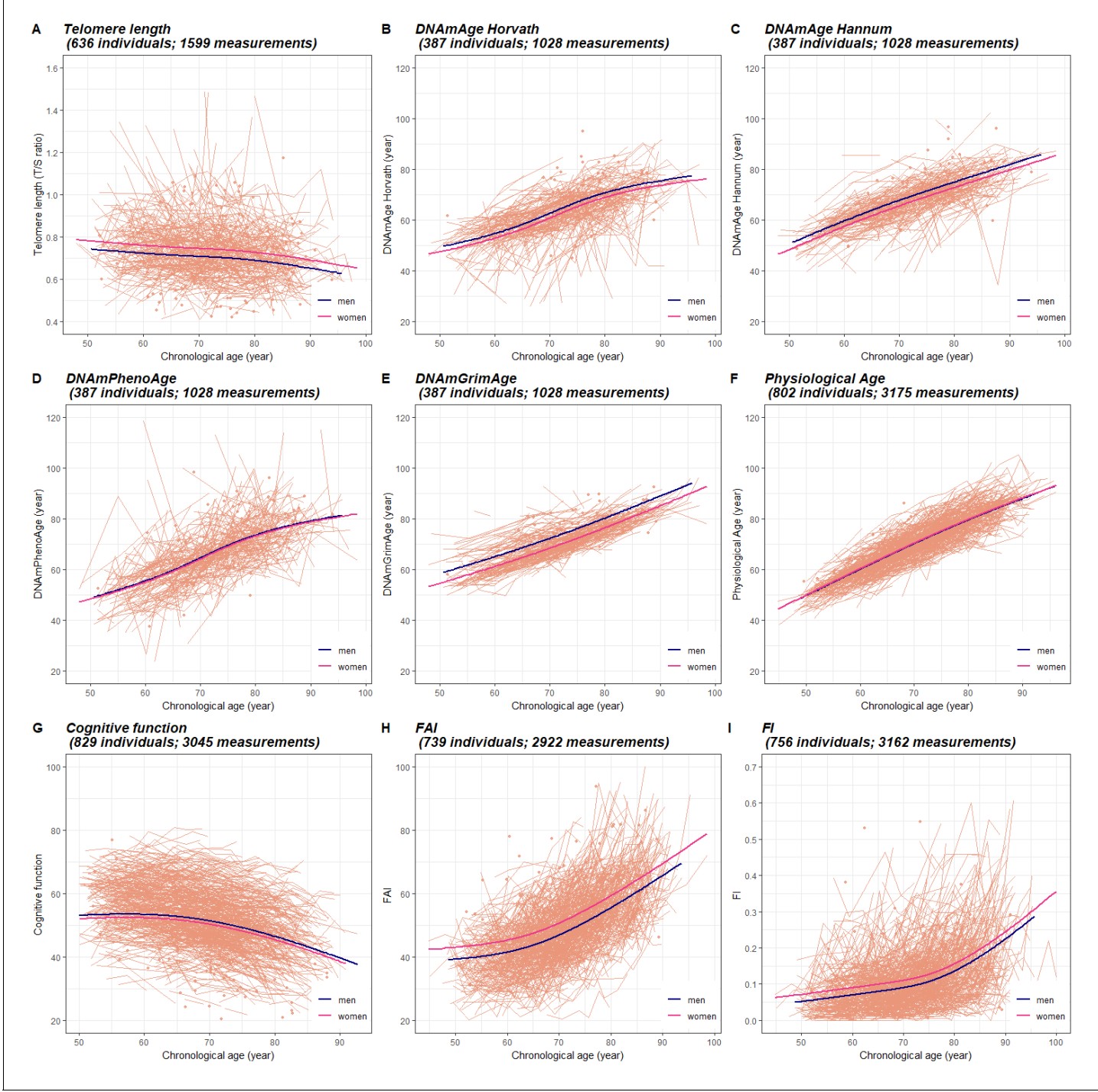

**Figure 1.** Longitudinal trajectories of BAs in 845 individuals (3973 measurements) with information on at least one BA. A total of 3973 repeated measurements assessed from 845 individuals were included to estimate the longitudinal trajectories of BAs. For each BA estimation, the numbers of available individuals and measurements varied and were specified in the heading of each panel. Longitudinal changes in BAs were modeled as functions of CA (as a natural spline with three degrees of freedom) and sex, with random effects at the individual and twin-pair levels (mixed models). Both individual-level BAs and population BA means over CA in men and women are presented in Panel (A-I). BA measurements were presented as orange dots, lines or broken lines when one, two, or more than two measurements were assessed for a given individual. Average changes of BAs with age in the study population were indicated by smooth lines (blue for men and pink for women). The longitudinal growth of the three functional BAs (cognitive function, FAI, and FI) show an accelerated rate of change around the age of 70 (Panel J-I), whereas the other BAs exhibit relatively linear trajecotries over the age span (Panel A-F). BA, biological age; DNAmAge, DNA methylation age estimator; FAI, functional aging index; FI, frailty index. The online version of this article includes the following figure supplement(s) for figure 1:

*Figure 1 continued on next page*

*Figure 1 continued*

**Figure supplement 1.** Longitudinal trajectory of physiological age with sex interaction term introduced to the mixed model.

the difference was mainly observed at the end of the CA spectrum (i.e., before the age of 50 and after the age of 85). (*Figure 1—figure supplement 1*).

## Correlations of BAs

BAs were broadly categorized into four groups according to the main biological structural levels where the BA measurements were implemented, that is telomere length, DNAmAges, physiological age, and functional BAs (Panel A in *Figure 2*). We estimated correlation coefficients between BAs and BA residuals and adjusted for repeated measurements using the rmcorr function in R (*Figure 2* and *Supplementary file 1E*) (*Bakdash and Marusich, 2017*). Using all complete measurements, we estimated repeated measures correlation (rmcorr, 22) between age and all nine BAs, which accounts for non-independence among observations and captures the common intra-individual association between age and each of the BAs in turn. Telomere length showed low correlations with both CA and BAs (absolute correlation coefficients $\leq 0.16$); the other BAs were correlated with CA and each other to varying degrees, with absolute correlation coefficients ranging from 0.24 to 0.87. As indicated by the correlation matrices of BAs and BA residuals (Panel B and C in *Figure 2*), after regressing out CA from BAs, most of the original correlations were attenuated and moderate correlations remained only between Horvath and Hannum DNAmAges (r = 0.35), cognitive function and FAI (r = −0.32), and FAI and FI (r = 0.31). Correlation patterns of BAs and BA residuals in men and women were similar to those in the whole population. (Data not shown).

Next, we transformed correlation coefficients into Euclidean distances and then performed hierarchical cluster analysis. The same types of BAs, that is methylation BAs and functional BAs, tended to be more closely related. GrimAge and PhenoAge, however, were somewhat separated from the other two DNAmAges, especially using BA residuals. (*Figure 2—figure supplement 1*).

## BAs and the risk of all-cause mortality

To study the association between BAs and all-cause mortality, we used Cox regression models with attained age being time scale and accounted for left truncation and right censoring. To make the

**Table 3.** Characteristics of baseline (first available) measurements in 845 individuals with information on at least one BA.

| | Baseline measurements in all individuals | Baseline measurements in individuals with corresponding BAs | | | | | |
|---|---|---|---|---|---|---|---|
| | | Telomere length (T/S ratio) | DNAmAge (year) | Physiological age (year) | Cognitive function | FAI | FI |
| Number of participants | 845 | 636 | 387 | 802 | 829 | 739 | 756 |
| Women (%) | 59.5 | 58.5 | 59.9 | 58.9 | 59.6 | 59 | 59.4 |
| Above primary education (%) | 42.2 | 45.4 | 45.5 | 41.8 | 42.6 | 43.7 | 43.3 |
| BMI (kg/m$^2$) | 25.7 (3.9) | 26.3 (4.1) | 26.3 (4.3) | 25.6 (3.9) | 25.7 (3.9) | 25.8 (4) | 25.8 (4.1) |
| Current and ex-smokers (%) | 25.0 | 22.2 | 21.7 | 24.1 | 25.5 | 24.1 | 23.7 |
| Age (year) | 63.6 (8.6) | 68.8 (9.6) | 69 (9.6) | 64.5 (8.9) | 63.7 (8.3) | 65.3 (9.2) | 65.5 (9.4) |
| BA | | 0.73 (0.17) | 60.4 (11.0) 65.2 (10.1) 63.8 (13.61) 69.4 (8.5) | 64.7 (10.3) | 51.5 (10.4) | 48.3 (11.4) | 0.10 (0.08) |
| Number of measurements | | 2.5 (1.3) | 2.7 (1.3) | 4.0 (2.2) | 3.7 (2.1) | 4.0 (2.4) | 4.2 (2.5) |

Values are means (standard deviations; SDs) unless stated otherwise.

Values of 'BA' in the 'DNAmAge' column refer to four different types of DNAmAge: Horvath, Hannum, PhenoAge, and GrimAge, respectively.

BA, biological age; IPT, In-person testing; DNAmAge, DNA methylation age estimator; FAI, functional aging index; FI, frailty index; BMI, body mass index.

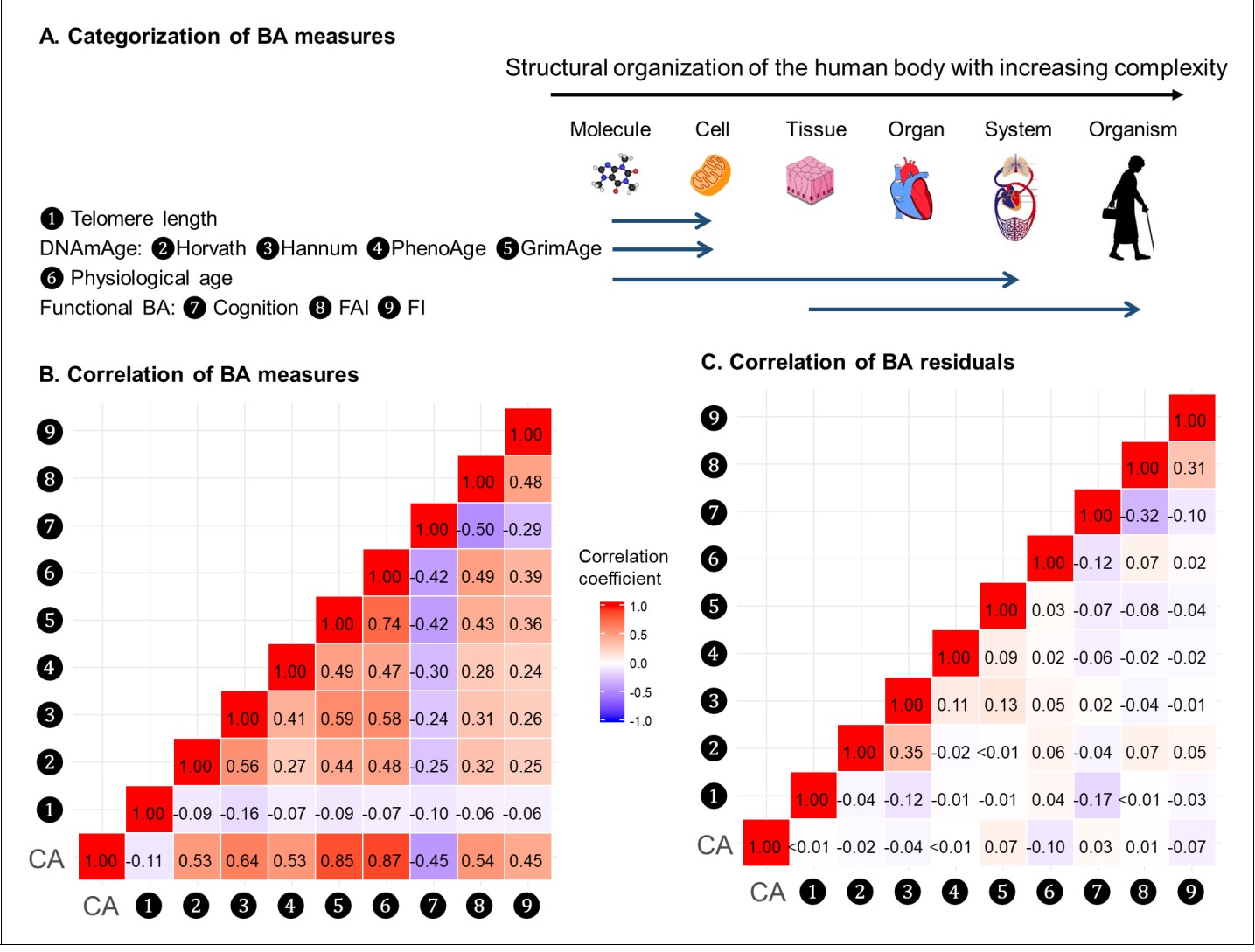

**Figure 2.** Correlations of BAs in 288 individuals (612 complete measurements). A total of 612 complete measurements assessed from 288 individuals were included to estimate the correlations of BAs. BAs were broadly categorized into four groups according to the main biological structural levels where the BA measurements were implemented (Panel A). We estimated the repeated-measure correlation coefficients between BAs and between BA residuals and illustrated the correlation coefficients in heat maps (Panel B-C). Red and blue tiles represented positive and negative correlations, respectively; color density indicated the magnitude of correlation coefficients. All BAs were correlated to varying degrees (Panel B). After regressing out CA from BAs, most of the original correlations were attenuated (Panel C). BA, biological age; DNAmAge, DNA methylation age estimator; FAI, functional aging index; FI, frailty index; CA, chronological age.

The online version of this article includes the following figure supplement(s) for figure 2:

**Figure supplement 1.** Hierarchical clustering of BA in 288 individuals (612 complete measurements).

hazard ratios (HR) comparable, we rescaled all BAs and BA residuals by their standard deviation (SD), so that HRs express changes in mortality risk due to a one-SD increment of the respective BA or BA residual. Fully adjusted models controlled for sex, educational attainment, smoking status, and BMI. All models were stratified by participants' birth year (categorized in 10 year intervals into five groups) and implicitly adjusted for CA through the choice of attained age as the underlying time scale. When repeated measurements were available, only the first BA assessment was included in the survival analysis and referred to as the baseline BA.

First, we looked at the effect of baseline BAs separately (i.e., one-BA models). During a median follow-up time of more than 15 years, we observed statistically significant BA-mortality associations for DNAmAges (GrimAge, PhenoAge, and Horvath) and functional BAs (FI, FAI, and cognitive

function), where a one-SD increase of BAcorresponded to a change in mortality risk by 39, 26, 17, 32, 27, and 15%, respectively. Increased mortality risks were also found with changes in Hannum DNAmAges and physiological age, although the effects were generally weaker and came with a slightly higher level of imprecision, with HRs (95% CI) being 1.17 (0.98, 1.40) and 1.13 (0.97, 1.31). We found no evidence for telomere length as associated with mortality risk. (*Table 4*) The corresponding HRs for the BA residuals were comparable to BAs (*Supplementary file 1F*).

Second, we explored the baseline BAs in relation to mortality risk by taking all BAs into consideration simultaneously in 288 individuals with complete measurements (i.e., nine-BA models). To avoid biased estimation due to collinearity (as all nine BAs were correlated to varying extents), BA residuals instead of BAs were used. During a median follow-up time of 16.5 years, we observed that three BA residuals, Horvath DNAmAge, DNAmGrimAge, and FI, out of nine were associated with the risk of all-cause mortality independently of all other BAs. A one-SD increment of Horvath DNAmAge, DNAmGrimAge and FI residuals was associated with a 31%, 43% and, 58% higher risk of death, respectively, independent of the level of CA, all other BA residuals and common risk factors. The HRs for the other DNAmAges (Hannum and PhenoAge), physiological age, and the other functional ages (cognitive function and FAI) were attenuated towards one after controlling for all BAs in the same model (*Table 5*).

### Sensitivity analyses

We replicated the first part of the survival analyses (i.e., one-BA models) in subgroups classified by sex, age group, and smoking status, as well as in the sub-cohort of individuals with complete measurements. In the subgroup analyses, we observed evidence for potential effect modification. The associations of BAs with mortality risk were generally stronger in women (except for Horvath DNAmAge, and physiological age), more pronounced in the younger individuals (except for Horvath DNAmAge, physiological age and cognitive function), and a bit stronger in current or ex-smokers (for Horvath DNAmAge and DNAmGrimAge) compared to those in men, older individuals, and non-smokers, respectively. (*Figure 3* and *Supplementary file 1G–I*) Furthermore, as with the results observed in the whole population, we found the same general pattern of BA associations with mortality in the sub-cohort with complete measurements. (*Supplementary file 1J*) We also adjusted for the presence of the previous diseases, including heart failure, stroke, diabetes, and cancer in the survival model and the observed results were largely unchanged. (*Supplementary file 1K*).

**Table 4.** Survival analyses of baseline (first available) BAs with the risk of all-cause mortality in 845 individuals (one-BA models).

| BAs | Number of individuals | Number of deaths | Median follow-up time (year) | Model 1 | Model 2 |
|---|---|---|---|---|---|
| Telomere length | 636 | 389 | 15.8 | 0.96 (0.87, 1.06) | 1.01 (0.92, 1.11) |
| DNAmAge (Horvath) | 387 | 240 | 16.1 | 1.14 (1.00, 1.32) | 1.17 (1.01, 1.36) |
| DNAmAge (Hannum) | 387 | 240 | 16.1 | 1.24 (1.05, 1.46) | 1.17 (0.98, 1.40) |
| DNAmPhenoAge | 387 | 240 | 16.1 | 1.22 (1.06, 1.40) | 1.26 (1.08, 1.47) |
| DNAmGrimAge | 387 | 240 | 16.1 | 1.49 (1.18, 1.89) | 1.39 (1.11, 1.75) |
| Physiological age | 802 | 543 | 18.7 | 1.12 (0.98, 1.29) | 1.13 (0.97, 1.31) |
| Cognitive function | 829 | 570 | 19.2 | 0.83 (0.75, 0.91) | 0.85 (0.76, 0.94) |
| FAI | 739 | 481 | 17.9 | 1.21 (1.06, 1.38) | 1.27 (1.10, 1.47) |
| FI | 756 | 498 | 17.7 | 1.28 (1.15, 1.43) | 1.32 (1.18, 1.48) |

Values are Hazard Ratios (95% Confidence Interval) [HR (95% CI)] unless stated otherwise.

HRs (95%CIs) in each column refer to the relative risks associated with one-SD increase in the level of BA of nine different models with one corresponding BA being the predictor of the mortality risk. Model 1 is the uni-variate survival model with only one BA taken into account. Model 2 is the multi-variate survival model, in which common risk factors (sex, education attainment, smoking status, and BMI) were additionally adjusted for on the basis of Model 1. All models were stratified by participants' birth year (in 10-year interval). Attained age was used as the time-scale and thus age was inherently adjusted for.

BA, biological age; DNAmAge, DNA methylation age estimator; FAI, functional aging index; FI, frailty index.

**Table 5.** Survival analyses of baseline (first available) BA residuals with the risk of all-cause mortality in 288 individuals with complete measurements (nine-BA models).

| BA residuals | Model 1 | Model 2 |
| --- | --- | --- |
| Telomere length | 0.98 (0.86, 1.12) | 1.03 (0.89, 1.19) |
| DNAmAge (Horvath) | 1.22 (1.01, 1.48) | 1.31 (1.08, 1.58) |
| DNAmAge (Hannum) | 1.05 (0.88, 1.26) | 1.03 (0.83, 1.28) |
| DNAmPhenoAge | 1.08 (0.87, 1.33) | 1.13 (0.91, 1.40) |
| DNAmGrimAge | 1.44 (1.19, 1.74) | 1.43 (1.11, 1.84) |
| Physiological age | 0.99 (0.88, 1.12) | 1.01 (0.87, 1.18) |
| Cognitive function | 0.92 (0.76, 1.12) | 1.01 (0.85, 1.20) |
| FAI | 1.08 (0.90, 1.29) | 1.04 (0.86, 1.27) |
| FI | 1.46 (1.24, 1.72) | 1.58 (1.32, 1.89) |

During a median follow-up time of 16.5 years, 151 deaths were documented among 288 individuals.

Values are Hazard Ratios (95% Confidence Interval) [HR (95% CI)] unless stated otherwise.

HRs (95%CIs) in each column refer to relative risks associated with one-SD increase in the level of BA residual from one multi-variate model with all BA residuals being the predictors of the risk of mortality simultaneously. Model one took eight BA residuals into account. Model two additionally adjusted for common risk factors (sex, education attainment, smoking status, and BMI) on the basis of Model 1. All models were stratified by participants' birth year (in 10 year interval). Attained age was used as the time-scale and thus age was inherently adjusted for.

BA, biological age; DNAmAge, DNA methylation age estimator; FAI, functional aging index; FI, frailty index.

## Discussion

The present study investigated nine different BAs in a middle and old-aged Swedish population-based cohort. We observed longitudinal growth of functional BAs accelerated around age 70 and sex differences in the average levels of BAs across the age spectrum. All BAs were correlated to varying degrees and the correlation coefficients were attenuated after adjusting for CA. Of nine BAs, four DNAmAges, physiological age, and three functional BAs were associated with the risk of mortality individually, with the strongest effect size observed for DNAmGrimAge and FI. In particular, Horvath DNAmAge, DNAmGrimAge, and FI remained predictive of mortality in a multivariable model including all BAs.

### Longitudinal trajectories of BAs

As biological aging does not proceed at a constant pace across time, monitoring longitudinal trajectories of BAs using repeated assessments to depict the characteristics of the aging course is of great importance. We observed an accelerated change in the aging pace in functional BAs, which are in parallel with previous studies focused on trajectories of cognition, physical function, and frailty (*Harris et al., 2016*; *Stow et al., 2018*; *Zaninotto et al., 2018*). This result indicates that an aged population may expect deteriorations of their functional performance to speed up as their age approach 70 years. In particular, we observed that the aging acceleration of cognition (cognitive function) and physical function (FAI) seemed to take place before age 70, while frailty (FI) went up at an increased rate after age 75. This evidence suggests that accelerated cognitive and physical dysfunctions are likely to arise prior to frailty's, and corresponding interventions and screenings could aim at people of different ages accordingly.

In contrast to functional BAs, DNAmAges and physiological age accrued at a relatively constant rate. Due to their mode of construction, DNAmAges and physiological age were transformed into an age-calibrated scale in the unit of year. Thus, these BAs allow individuals to perceive an intuitive impression on their biological aging status by comparing BA with CA and have the potential to serve as a straightforward-to-use metric in the geriatric practice. However, in order to be clinically meaningful, they need to be validated on the individual level data as well. In addition, a drawback of this mode of construction is that it largely explains the linear shapes of the BA trajectories, making longitudinal BA characteristics less informative as the changes are constant. Another feature of physiological age is the strong correlation with CA; hence, it allows limited biology perception in prediction

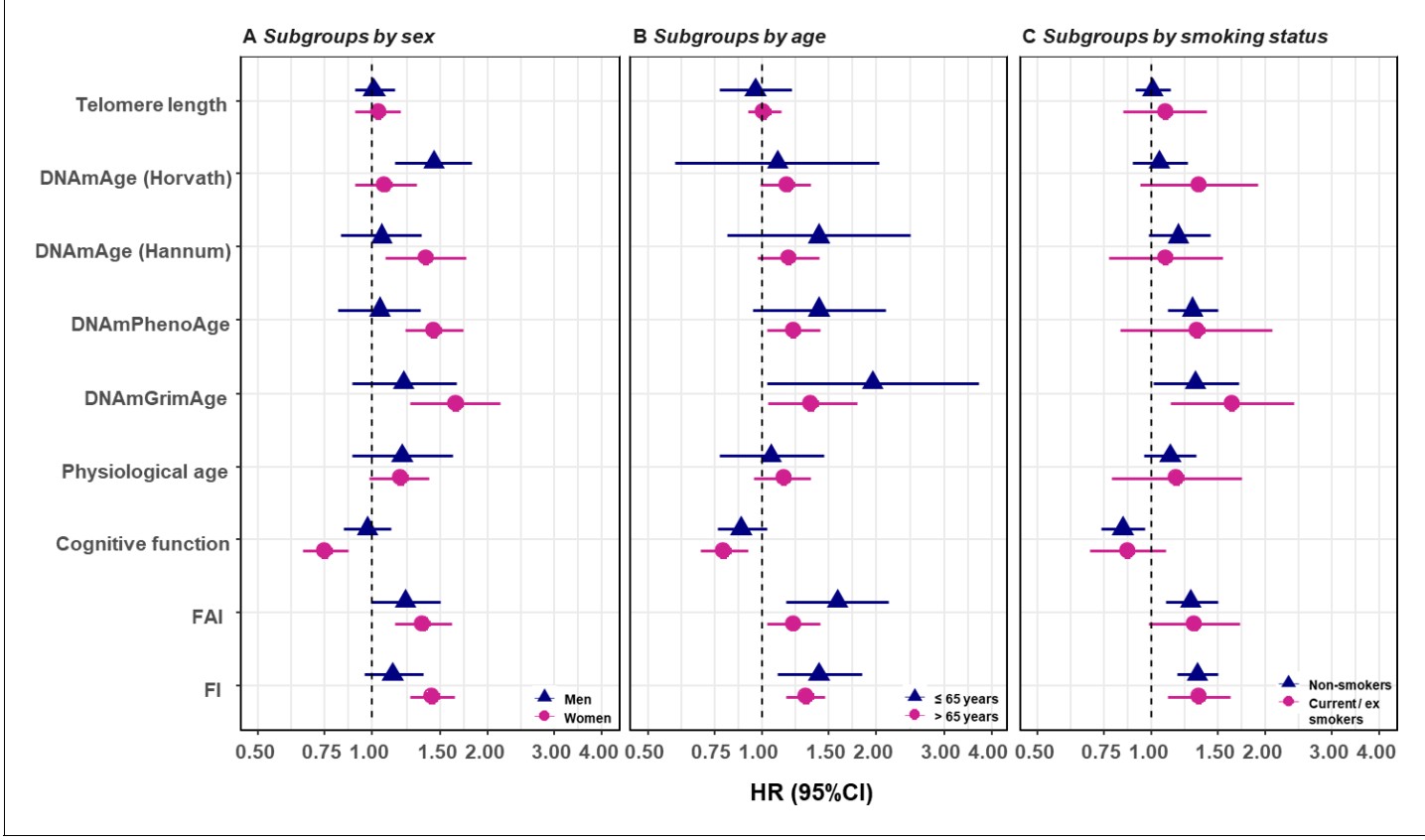

**Figure 3.** Survival analyses of baseline BAs with the risk of all-cause mortality in subgroups classified by sex, baseline smoking status, and baseline age (one-BA models). A total of 845 individuals were included to estimate the mortality associations of BAs in subgroups. The numbers of individuals in each subgroups were specified in the *Supplementary file 1G–I*. We used Cox regression models to estimate the change in mortality risk associated with a one-SD increment of the respective BA at baseline assessment (one-BA models). All models controlled for sex, educational attainment, smoking status, and BMI, stratified by participants' birth year, and accounted for left truncation and right censoring. Attained age was used as the time-scale and thus age was inherently adjusted for. BA-mortality associations by were illustrated in the forest plot (Panel A-C), in which points and horizontal lines denoted HRs (95%CIs) and point shapes and colors represented subgroups. The associations of BAs with mortality risk were generally stronger in women (except for Horvath DNAmAge and physiological age), more pronounced in the younger individuals (except for Horvath DNAmAge, physiological age and cognitive function), and a bit stronger in current or ex- smokers (for Horvath DNAmAge and DNAmGrimAge). BA, biological age; DNAmAge, DNA methylation age estimator; FAI, functional aging index; FI, frailty index; CA, chronological age; HRs (95%CIs), hazard Ratio (95% Confidence Interval).

models on top of CA. This is again due to its mode of construction using a composite score of several biomarkers where each one, by definition, strongly correlates with CA. To improve construction models for BA, composite scores should balance the reliance on biological aspects and weight on CA.

Sex differences in aging are observed at multiple biological levels (*Oksuzyan et al., 2008*; *Ostan et al., 2016*). Most notably, women tend to present lower mortality but higher morbidity rates, especially at advanced ages, compared to men of the same age. We found that, compared to men, women presented lower molecular ages (telomere length and methylation age estimators) but higher functional ages (FAI and FI) across the age spectrum, indicating different medical needs are likely to be encountered for men and women. Meanwhile, we observed no evidence for shape differences in the BA trajectories (except for physiological age) by sex. This suggests that aging proceeded at a similar rate in men and women since midlife onwards. For physiological age, the shape difference was mainly observed at the end of the age spectrum and could be influenced by methodological issues as the construction of physiological age was done differently in men and women. Putting our results in the perspective of previous studies, we confirm earlier findings in women, such as better telomere status and worse frailty (*Mu et al., 2014*; *Bartley et al., 2016*). One possible

explanation for the sex difference in FI is that men and women tend to present different self-perception on aging-related performances due to social and cultural considerations when self-reported data are used (*Ostan et al., 2016*). However, cognitive function and FAI incorporate objective measurements tested by trained nurses and both of them exhibited sex differences in the same direction. Hence, sex differences in the functional BA spectrum are not, or at least not exclusively, driven by social and cultural factors. Therefore, we suggest that sex differences should always be accounted for in studies of aging and, if possible, presenting stratified analyses should be encouraged.

## Correlations of BAs

Aging is a complex process proceeding through interconnected biological mechanisms. BAs, in essence, are underlain by different sets of these mechanisms. Correlations of BAs could provide an idea of how one BA changes in accordance with another. We observed moderate to strong correlations between CA and all BAs, with the exception of telomere length, which was only weakly correlated with CA and the other BAs. These results are in line with previous research observing similar correlation pattern for 11 BAs, as well as low correlations between telomere length and other types of BA (*Belsky et al., 2018*). As some degree of correlation between BAs is expected simply due to their relation with CA, we also investigated correlations between BA residuals, which by construction minimize correlation with CA. We found clear attenuation of the correlations between different types of BA residuals, that is methylation BAs and functional BAs as well as physiological age and telomere length. In other words, among individuals with the same CA, molecular BAs and functional BAs are only weakly associated. We interpret this as an indication that these categories largely reflect different aspects of the aging process.

After regressing out the effect of CA, we still observed weak to moderate correlations between DNAmAges. These DNAmAges were initially trained to capture methylome differences associated with various aging-related phenotypes, namely CA for Horvath and Hannum, Phenotypic Age for PhenoAge, and mortality risk for GrimAge (*Hannum et al., 2013*; *Horvath, 2013*; *Levine et al., 2018*; *Lu et al., 2019*). Employing correlated surrogates and measuring the same aging hallmark of epigenetic alteration in the development of these DNAmAges could partly explain the correlations observed among these residuals. Hierarchical clustering suggested GrimAge and PhenoAge were somewhat separated from the other two DNAmAges, which is expected as GrimAge and PhenoAge incorporate additional information on top of CA in the development of their BA algorithms. In addition, Horvath and GrimAge were the least correlated DNAmAge residuals, indicating that these two BAs captured uncorrelated methylome information to a larger degree than other DNAmAges did and might have the potential to inform aging-related outcomes independently (as illustrated in the results of nine-BA survival analyses).

Functional BA indicators included cognitive function, FAI, and FI, which quantify cognitive, physical, and frailty-related functioning. Intriguingly, the present analysis found comparable levels of correlation for these BAs as well as for their residuals, suggesting that most of these correlations were not simply driven by CA. Previous studies have also reported correlations between physical function, cognitive function, and health-related quality of life (*Kim, 2016*). As functional BAs measure complex phenotypes resulting from intricate biological mechanisms, further investigations are needed to disentangle the mechanisms underlying both the age-dependent and, in particular, the age-independent correlations.

## BA-mortality associations

A majority of the included BAs are well-established aging indicators and were found to be associated with mortality in previous studies (*Hannum et al., 2013*; *Levine, 2013*; *Shamliyan et al., 2013*; *Wang et al., 2018a*). Our results were comparable to the current evidence. We first examined the baseline value of individual BAs in relation to the risk of death and found associations for all BAs except for telomere length, with DNAmGrimAge and FI demonstrating the strongest relationship. We next examined BA-mortality associations with all BAs included in the same model and observed significant evidence for Horvath DNAmAge, DNAmGrimAge, and FI, suggesting they were complementary in the prediction of mortality. In other words, Horvath DNAmAge, GrimAge and FI were likely to capture aging-related information that was both informative in the prediction of mortality risk and independent of other BAs. DNAmAges (Hannum and PhenoAge) and functional BAs

(cognitive function and FAI) were likely to reflect some mortality-associated aging aspects that were shared and/or correlated with other BAs, as the corresponding HRs were attenuated to almost one when all BAs were included in the model.

Few studies have focused on the comparison of the predictive aspect of BAs. *Kim et al. (2017)* found FI to outperform methylation age estimators in survival models by comparing only two types of BAs, frailty, and DNAmAge (Horvath). *Zhang et al. (2017)* developed a methylation-based mortality score and observed a stronger mortality association than the FI-mortality relationship. Furthermore, Murabito and colleagues compared clinical age, inflammatory age, and DNAmAge (Horvath and Hannum) and concluded that they were complementary in predicting risk for mortality (*Murabito et al., 2018*). Our study thus supports these findings, both in the way that mortality-oriented DNAmAges and FI were strongly associated with mortality, and in that certain types of BAs could reflect mortality risk independently. In particular, we add information by including other BAs, especially physiological age, a newly trained DNAmAge (GrimAge) and functional BAs.

Physiological age is a composite biomarker of aging, developed from 10 blood and clinical markers. Levine et al. applied a similar method to develop a BA indicator and found it associated with mortality beyond CA in a large US population (*Levine, 2013*; *Klemera and Doubal, 2006*). We also observed that physiological age predicted mortality risk independently of age, albeit weaker and at a moderate significance level. Despite its weak mortality association, physiological age presents some benefits with respect to the physiological interpretation and the search for biological mechanism in future explorations, as the component biomarkers contributed to the development of physiological age were explicitly specified.

DNAmGrimAge is derived via a two-step development method, in which methylation data were first used to predict a set of biomarkers (plasma proteins and smoking pack-year), and the methylation-based biomarkers were then used in the prediction of mortality risk. As a result, the DNAmGrimAge is explicitly trained to be a mortality predictor; it also includes the largest number of genomic sites compared with the other three DNAmAges, and allows the methylation level of a single CpG site to contribute to mortality risk via different intermediate biomarkers, as the methylation-based biomarkers used somewhat overlapping CpG sites in the two-step training phase. *Lu et al. (2019)* reported the advantage of predicting mortality over other existing DNAmAges in a US population. These results point to the potential of utilizing methylation information to reflect complex aging phenotypes.

The FI is developed using clinical information and is likely to capture an organismal perspective on aging that is closer to the clinical end-point of death compared to molecular-based BAs. In addition, the FI incorporates information in multiple health domains, in contrast to the other two functional BAs, cognitive function, and FAI, which exclusively measure cognitive or physical performance. The comprehensive nature of the FI could partly explain the strong and robust mortality association observed here and in previous studies (*Kojima et al., 2018*). Functional BAs, especially FI, measure not only the biological aging process, but also social and psychological wellbeing. Consequently, disentangling the underlying biological mechanisms from the functional measures would be challenging, which impairs the ability of functional BAs to explain the innate biological aging process. Specifically, it is difficult to implement functional BA measurements in animal models, whereas molecular BAs can be easily investigated in animal studies. Additionally, functional BAs in humans tend to show less heterogeneity among the young, making them less ideal instruments to assess the aging process among young adults, compared to using molecular BAs. This is especially true when it comes to FAI as it only takes four functioning indicators into account. However, FI considers a wide range of health deficits (42 in our case) and provides relatively precise scales to differentiate individuals with lower functioning scores. Furthermore, since biological, psychological, and social factors all contribute to the health status of human beings, the comprehensive nature of functional BAs also comes with important advantages for predicting aging-related outcomes, underlining the complementary nature of molecular and functional BAs.

## Sensitivity analysis

In the subgroup analysis, we found moderate effect modification from age and smoking status. A previous study in a large Swedish cohort from our research group also observed that the effect size of FI with mortality was attenuated with increasing age (*Li et al., 2019*). DNAmGrimAge includes smoking (pack-year) associated CpG sites in its establishment (*Lu et al., 2019*), which could partly

explain a stronger association observed in smokers than non-smokers as it may capture smoking-based deterioration at the molecular level. In addition, the present results found a majority of the BA-mortality associations to be stronger in women compared to men, suggesting sex differences should be acknowledged in making use of BAs to inform clinically relevant decisions.

It is noteworthy that in comparing the strength of relative risk of mortality related to each BA, we were not aiming to propose any superior BA, as mortality association alone is by no means the gold standard for a qualified BA in aging research. Evidence exists that some BAs have advantages in predicting other aging-related outcomes such as entry into care (e.g., FAI) (*Finkel et al., 2019*), which may be more relevant for developing policies and interventions.

## Strengths and limitations

The present study has several strengths. First, we measured nine BAs in the same population from the molecular to the organismal level, which provided us an opportunity to examine BA characteristics from several molecular and functional aspects. Second, the BAs were assessed in a longitudinal manner to allow for descriptive and correlational analyses considering changes over time. Third, through the linkage to the Swedish national registers, we were able to keep track of study participants' vital status for nearly 20 years, thus being able to predict the risk of death in relation to BA status prior to the occurrence of most aging-related outcomes.

Several limitations should be taken into consideration when interpreting the present findings. First, only individuals with complete measurements were included in the nine-BA survival models. Complete observations may not be a representative subgroup of the entire study population and lead to selection bias. However, IPT occasions with incomplete BAs were mostly due to administrative reasons, such as methylation data were only assessed in five waves since IPT3, and these factors are likely not related to subject-specific features. In addition, we replicated the one-BA survival models in individuals with complete measurements, and the results did not show major discrepancies. Second, the birth year of SATSA participants spanned a long period of time, from the year 1900 to 1948, thus raising concerns for cohort effects. During this time, improved medical conditions or emergent public health events could affect both BA and mortality risk and cannot be controlled merely through the age scale in the present analysis. All survival models were stratified by participants' birth year in 10 year intervals to in part alleviate bias due to the cohort effect.

In summary, biological age is a construct built on molecular, cellular, and functional aspects of an individual´s health status and has the potential to change the way risk prediction is performed in the clinic. In the era of personalized medicine, individual health assessments are warranted and should rely on better methods than simply using the CA. In this study, we present the most comprehensive analysis of biological age in humans to date. We explain their correlations and longitudinal patterns. We highlight the sex-specific effects and point to the importance of providing sex-specific estimates in aging investigations and acknowledging sex-specific care in geriatric practices. Furthermore, we found BAs had the potential to provide mortality-relevant information independently of CA and independently of other types of BAs. Further studies are needed to investigate if the present findings could be extrapolated to a younger population, what the specific mechanisms underlying the BA correlations are, and possible modification effects on and between BAs.

## Materials and methods

### Study population

The Swedish Adoption/Twin Study of Aging (SATSA) was a population-based study consisting of reared apart and reared together twin pairs (*Finkel and Pedersen, 2004*). In-person testing (IPT) in SATSA was initiated in 1986 and nine complete waves of in-person testing were conducted through 2014 (IPT1 to IPT10, except for IPT four where only telephone interviews were performed), during which the information of a maximum of nine BAs, including telomere length, four DNAmAges (Horvath's, Hannum's, PhenoAge, and GrimAge), physiological age, and three functional ages (cognitive function, FAI, and FI) were available. In total, 859 individuals participated in at least one IPT wave in SATSA, and 846 individuals who had at least one BAs assessed once since IPT1 to IPT 10 (except for IPT4) and the information of vital status through linkage between the Swedish Twin Registry (STR)

and the Swedish Population Register were included in the present study. The number of individuals and measurements by IPT and BAs were detailed in *Table 1* and *Supplementary file 1A*.

## Assessment of biological age

### Telomere length

Telomere length was measured from DNA extracted from leukocytes in peripheral blood (*Berglund et al., 2016*). A quantitative polymerase chain reaction-based technique was carried out to compare the telomere sequence copy number in each participant's sample (T) to a single-copy reference gene from β-hemoglobin (S). The resulting relative length was represented as T/S ratio. Relative telomere length was further adjusted for batch effect, and 10 outliers (beyond Mean ±4*SD) were omitted in the present analyses.

### DNA methylation age estimator (DNAmAge)

Genome-wide methylation levels were measured from leukocytes using Illumina's Infinium Human-Methylation 450K BeadChip according to the manufacturer's protocol and quantified by beta-values (*Wang et al., 2018b*). DNAmAges of Horvath and Hannum versions incorporate methylation levels of 353 and 71 age-related CpGs trained from multiple sample types and blood sample accordingly through a penalized regression model (*Hannum et al., 2013*; *Horvath, 2013*). In contrast to DNA-mAges trained via regressing on age, the third version of methylation age estimator, PhenoAge, were trained on a composite clinical measure of phenotypic age, and eventually included 513 CpG sites (*Levine et al., 2018*). Further, GrimAge adopted a two-step development method, in which methylation data were used to predict a set of biomarkers (plasma proteins and smoking pack-year), and the methylation-predicted biomarkers were then used to predict mortality risk. As a result, a total number of 1030 CpG sites were taken into account (*Lu et al., 2019*). All DNAmAges were combined using penalized regression models and generated from the online DNA Methylation Age Calculator (*Horvath, 2019*).

### Physiological age

Physiological age considered a set of physiological biomarkers assessed from the immediate blood test, blood test in lab, urine strip test, and physical examination data that were available in all waves of IPT. First, we included one measurement for each individual to form a sub-sample in which one measurement was randomly selected when repeated measurements for a single individual were available. Pearson correlations were examined using measurements of age and candidate bio-markers in the sub-sample. As a result, nine and five eligible age-associated biomarkers (Pearson correlation >0.1) were included in the development of physiological age for men and women separately. *Supplementary file 1B* illustrates the biomarker-age correlations in detail. We then performed principal component analysis to created principal components (PCs) and applied a method proposed by *Klemera and Doubal (2006)* to combine CA and PCs into a single physiological age value in men and women separately using sub-sample. Second, we calculated PCs and physiological age for all available repeated measurements in men and women separately using loadings of bio-markers and weights of CA and PCs which were estimated from the sub-sample analysis.

### Cognitive function

Four cognitive domains were assessed through a battery of in-person cognition testing, including crystallized, fluid, memory, and perceptual speed abilities (*Reynolds et al., 2005*). Scores on all cognitive measures were recorded to percentage correct of the total possible points for each respective test. A general cognitive ability score was derived through the principal component analysis (PCA) of the tests. Component scoring coefficients from the first component extracted at IPT1, excluding demented individuals, were used to construct a cognitive function measure at the first and subsequent IPTs using test scores standardized to the mean and SD of each test at IPT1. T-score scaling (M = 50, SD = 10) was then applied to the components.

### Functional aging index (FAI)

Four types of specific functional measurements were taken into consideration in the development of FAI (*Finkel et al., 2019*). Vision and hearing were self-reported on a scale of 1 to 5 and combined to

create a measure of self-reported sensory ability. Muscle strength, walking speed time, and lung function were tested and recorded by trained nurses. The four indicators were standardized separately on the basis of the values from IPT two and then summed to create a composite score.

### Frailty index (FI)

FI was introduced to conceptualize the vulnerability of a given person to a range of age-related adverse outcomes. FI in SATSA was constructed from 42 self-reported health deficits, such as symptoms, diseases, disability, mood, and activities in daily living. FI was calculated as the ratio of the number of deficits presented in a given person to the total number of deficits considered in the study (n = 42 in SATSA). Details of FI items are described in *Supplementary file 1C* and elsewhere (*Jiang et al., 2017*).

### BA residuals

We constructed BA residuals by regressing out the CA-related part from respective BA. As BAs were assessed in a longitudinal manner and BA levels within twin pairs were assumed to be related due to shared familial factors, we adopted mixed models with fixed effects for sex and CA, the latter as a natural spline term with three degrees of freedom, to allow for non-linear relationships, and random intercepts at the twin-pair and subject level:

$$BA_{ijk} = \beta_0 + ns(CA_{ijk}, \beta_1, \beta_2, \beta_3) + \beta_4 * Sex_{ij} + \mu_0 i + \mu_0 ij + BAResidual_{ijk}$$

with $\beta$ and $\mu$ denoting fixed and random effects, i, j, k being indicators for twin pair, individual, and measurement, respectively, and $ns()$ representing a natural spline term with parameters as specified by the degrees of freedom. The resulting predicted residuals $BAResidual_{ijk}$ have thereby been adjusted for CA as well as systematic constant differences between twin pairs and individuals.

## Assessment of covariates

Sex, educational attainment, and baseline information of BMI, and smoking status were considered as covariates in the survival analyses. BMI was assessed through physical examination and other covariates were acquired through self-reported questionnaire data. Educational attainment was classified as primary education, lower secondary or vocational, upper secondary education, and tertiary education. BMI was derived as the body mass divided by the square of body height, expressed in units of $kg/m^2$. Smoking status was categorized as non-smokers, ex-smokers, and current smokers. Among the included individuals and measurements, 28 out of 846 individuals without education information were assigned into an unknown group, and another one and two individuals who missed the BMI and smoking status were imputed with the corresponding information collected from their nearest available IPTs.

Since the participants in the present analysis were born over a long period of time (1900 to 1948), we estimated the mortality associations with stratification on the calendar year of the participants' birthday to avoid confounding from cohort effects. Individuals' birth year was treated as a categorical variable with five groups in an interval of 10 years.

## Assessment of mortality

All-cause mortality data, including vital status and dates of deaths, were obtained from linkages between the STR and Swedish Population Register through the personal identification number assigned to all residents. All-cause mortality data were updated on August 16, 2018.

## Statistical methods

Characteristics of baseline measurements were presented as means (standard deviations, SDs) and proportions. Longitudinal changes of BAs were presented in plots with individual level measurements illustrated by dots, lines, and broken lines. Average trajectories of BAs were estimated through mixed linear regression models as described above, each of which included fixed effects of the intercept, CA, and sex, and a random effect of intercept at the individual and twin-pair level. P-values for sex effects were obtained from fixed effects within the mixed linear model. In addition, an interaction term between each BA and CA in natural spline was introduced to the model, and

P-values for sex interaction were determined by the likelihood ratio test comparing the models with and without the interaction term.

We estimated the repeated measures correlation coefficients between age and nine BAs using all complete measurements (*Bakdash and Marusich, 2017*). The correlation analyses were then replicated for CA and nine BA residuals to examine the correlations between the parts of BA residuals. We then transformed correlation coefficients to scaled squared Euclidean distances and performed hierarchical cluster analysis on BAs and BA residuals via Ward's method.

We used the Cox regression model to estimate the association between baseline BAs and the risk of all-cause mortality. When repeated measurements were available, only the first BA assessment was included and referred to as the baseline BA. In all models, we used attained age as the underlying time scale and the five groups of birth year as strata. Left truncation and right censoring were accounted for in the estimations; that is individuals entering into the cohort is conditional on their survival at baseline ages, and follow-up time may stop before death occurs. Follow-up time started from the first available measurement (baseline measurement) and ended at the date when they died or were censored on August 16, 2018. In addition, robust standard errors were introduced to adjust for relatedness within twin pairs and subjects. To achieve a direct comparison of BA effects in relation to mortality risk, we standardized all nine BAs to the mean of zero and SD of one across all available measurements and replicated the transformation in the BA residuals, such that the estimated hazard ratios (HRs) could be interpreted as the relative risk of death associated with a one-SD increase in the level of BA or BA residual. In the first part of the survival analyses, all models accounted for only one BA (one-BA models). For those who had their BAs assessed for multiple times, the first available measurement was treated as the baseline measurement. For each BA, we estimated the association of BAs and BA residuals with mortality risk using two models, a univariate model and a multivariate model with sex, education, BMI, and smoking status as additional covariates. In the second part of the survival analyses, all nine BAs were accounted for altogether in the same survival model (multi-BA models). Similarly, two models with or without adjustment for common risk factors were estimated. Only BA residuals were taken into consideration in the second part of survival analyses to avoid biased results caused by collinearity within BAs.

P values were two-sided, and statistical significance was defined as p<0.05. All analyses were conducted using Stata 15.1 and R 3.6.0.

## Additional information

### Funding

| Funder | Grant reference number | Author |
|---|---|---|
| National Institutes of Health | R01 AG04563 | Nancy L Pedersen |
| MacArthur Foundation | Research Network on Successful Aging | Nancy L Pedersen |
| Swedish Research Council for Health, Working Life and Welfare | 97:0147:1B | Nancy L Pedersen |
| Swedish Research Council | 825-2007-7460 | Nancy L Pedersen |
| Swedish Research Council | 2015-03255 | Sara Hägg |
| Karolinska Institutet | Foundation for Geriatric Diseases | Sara Hägg |
| Magnus Bergvall Foundation | | Sara Hagg |
| Karolinska Institutet | Strategic Research Program in Epidemiology | Sara Hagg |
| King Gustaf V and Queen Victoria's Foundation of Freemasons | | Sara Hagg |
| China Scholarship Council | | Xia Li |
| Swedish Research Council | 2017-00641 | Patrik KE Magnusson |

| Swedish Research Council | 2018-02077 | Juulia Jylhävä |
| National Institutes of Health | R01 AG10175 | Nancy L Pedersen |
| National Institutes of Health | R01 AG028555 | Nancy L Pedersen |
| Swedish Research Council | 825-2009-6141 | Nancy L Pedersen |
| Swedish Research Council | 521-2013-8689 | Nancy L Pedersen |
| Swedish Research Council | 2019-01272 | Sara Hägg |
| Swedish Research Council for Health, Working Life and Welfare | 2009-0795 | Nancy L Pedersen |
| Swedish Research Council for Health, Working Life and Welfare | 2013-2292 | Nancy L Pedersen |

The funders had no role in study design, data collection and interpretation, or the decision to submit the work for publication.

### Author contributions
Xia Li, Formal analysis, Methodology; Alexander Ploner, Patrik KE Magnusson, Juulia Jylhävä, Supervision, Methodology; Yunzhang Wang, Chandra Reynolds, Deborah Finkel, Methodology; Nancy L Pedersen, Funding acquisition, Methodology; Sara Hägg, Conceptualization, Supervision, Methodology

### Author ORCIDs
Xia Li (iD) https://orcid.org/0000-0003-1922-7152
Alexander Ploner (iD) http://orcid.org/0000-0002-5042-8326
Sara Hägg (iD) https://orcid.org/0000-0002-2452-1500

### Ethics
Human subjects: All participants provided informed consent. This study has received ethical approval from the Regional Ethics Review Board, Stockholm (Dnr 2016/1888-31/1).

### Decision letter and Author response
Decision letter https://doi.org/10.7554/eLife.51507.sa1
Author response https://doi.org/10.7554/eLife.51507.sa2

## Additional files

### Supplementary files
• Supplementary file 1. Additional information with respect to methods and results in detail. (**A**) Number of individuals with information on BAs by IPTs. (**B**) CA-biomarker correlations in the development of physiological age. (**C**) List of the 42 items included in the FI and their scoring. (**D**) Characteristics of baseline (first available) complete measurements in 288 individuals. (**E**) Repeated measures correlation coefficients of BAs in 288 individuals. (**F**) Survival analyses of baseline (first available) BA residuals with the risk of all-cause mortality in 845 individuals (one-BA models). (**G**) Survival analyses of baseline BAs with the risk of all-cause mortality in 845 individuals stratified by sex (one-BA models). (**H**) Survival analyses of baseline BAs with the risk of all-cause mortality in 845 individuals stratified by age group (one-BA models). (**I**) Survival analyses of baseline BAs with the risk of all-cause mortality in 845 individuals stratified by smoking status (one-BA models). (**J**) Survival analyses of baseline BAs with the risk of all-cause mortality in 288 individuals with complete measurements (one-BA models). (**K**) Survival analysis with additional adjustment for previous diseases.

• Supplementary file 2. 14 variables related to the current analysis, including anonymized personal- and twin-pair IDs, measurement number, sex, CA at measurement, and levels of nine BAs.

• Transparent reporting form

• Reporting standard 1. STROBE Statement—checklist of items that should be included in reports of observational studies.

## Data availability

Information on sex, CA, and nine BAs were listed in the Supplementary file 2. Besides, the SATSA cohort has been archived through NACDA program on aging (https://www.icpsr.umich.edu/icpsr-web/ICPSR/studies/3843) and detailed information on the study can be found on the Maelstrom Research platform (https://www.maelstrom-research.org/mica/individual-study/satsa). Data archiving is completed for IPT1-7 and is a work in progress for IPT8-10. In addition, all methylation array data are available in the Array Express database of EMBL-EBL (www.ebi.ac.uk/arrayexpress) under the accession number of E-MTAB-7309.

The following datasets were generated:

| Author(s) | Year | Dataset title | Dataset URL | Database and Identifier |
|---|---|---|---|---|
| Pedersen NL | 2015 | Swedish Adoption/Twin Study on Aging (SATSA), 1984, 1987, 1990, 1993, 2004, 2007, and 2010 | https://www.icpsr.umich.edu/icpsrweb/ICPSR/studies/3843 | ICPSR, 3843 |
| Yunzhang Wang | 2018 | DNA methylation of longitudinal samples from The Swedish Adoption/Twin Study of Aging (SATSA | https://www.ebi.ac.uk/arrayexpress/experiments/E-MTAB-7309/ | ArrayExpress, E-MTAB-7309 |

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
