## [Decision Letter]

**Acceptance summary:**

This is an important and timely study for the aging research field and more broadly for biomedical researchers. The study addresses the important question about validating promising human biomarkers of aging within a robust investigational design.

**Decision letter after peer review:**

Thank you for submitting your article entitled "Longitudinal trajectories, correlations and mortality associations of nine biological ages across 20-years follow-up" for further consideration by *eLife*. Your article has been reviewed by three peer reviewers, and the evaluation has been overseen by Diane Harper, as Reviewing Editor, and Eduardo Franco, as Senior Editor.

There are some issues that need to be addressed before acceptance, as outlined below:

Reviewer #1:

I think this is an important and timely study for the aging research field and more broadly biomedical researchers. The study addresses the important question about validating promising human biomarkers of aging. The study stands out in terms of its scope (profiling many different biomarkers ranging from DNA methylation to frailty index), its large sample size: longitudinal assessments from n=845 individuals, long follow up times, and its rigorous statistical approach (linear mixed modeling for adjusting for dependencies). It carefully evaluates the latest molecular biomarkers (e.g. recent methylation based biomarkers) relates them to a hard outcome: time to death. The sensitivity analysis illustrates robustness. The discussion seems to be fair and is grounded on the results. The statistical analysis appears to be sound.

Reviewer #2:

The advantages of this work are longitudinal (20-year) studies of several approaches to biological age estimation (nine measurements) at once, including data on mortality, in the same population.

Not sufficiently discussed is why a combination of these 9 marker groups was chosen? What are their mean absolute errors? How many persons have different longitudinal time points with all the nine? If such patients exist, is it possible to give individual correlations of different BAs with age?

Is it possible to conclude that longer telomeres or shorter DNAmAge are practically not associated with physiological indicators of functioning (clinical end-points)? Could the authors emphasise this aspect of the results in the Discussion?

Reviewer #3:

In this paper the authors assessed longitudinal measurements of multiple biological age in a population-based cohort from Sweden with long term follow-up. The dynamic changes of 4 types of 9 BAs over 20 years, mutual correlations and individual/joint associations with mortality of the BAs were explored. The study stands out in terms of longitudinal design, repeated measures of multiple types of BAs, and adequate statistical analysis. The results further confirmed the validity of multiple BAs as aging indicators and their potential complementary roles in aging process, and highlighted the variations with respect to gender. The following comments are mainly to improve the clarity.

1) Longitudinal changes of BAs over CA were investigated, whereas longitudinal changes of BAs in relation to mortality were not explored.

2) Follow-up time for cognitive function and physiological age (and even FAI and FI) were obviously longer than follow-up time of other BAs, it therefore may not be appropriate to compare the strength of different BAs with mortality given that only the first BA assessment was included in the survival analysis. What are the results if truncating follow-up time for different BAs comparable?

3) What are the health status of participants at baseline (first BA assessment), which may affect mortality risk and is suggested to be controlled in regression models.

4) Categories of BAs are suggested to be described explicitly in the Introduction.

---

## [Author Response]

Reviewer #2:The advantages of this work are longitudinal (20-year) studies of several approaches to biological age estimation (nine measurements) at once, including data on mortality, in the same population.

We thank the reviewer for this positive feedback and the following helpful comments to improve our work.

Not sufficiently discussed is why a combination of these 9 marker groups was chosen? What are their mean absolute errors?

a) The main objective of the present study is to investigate different types of biological ages (BAs) jointly. Therefore, we intended to include as many BAs as possible so that a range of BA types could be compared. As such, the nine BAs are selected mainly based on data availability and we have added the following sentence to the manuscript:

“Nine BAs were included in the present study based on data availability in SATSA.”

b) In Author response image 1, we have scaled BAs to standard deviation (SD) units one by one and calculated the mean absolute error (MAE) pair by pair as mean absolute error (MAE) pair by paΣi=1n|BA1i-BA2i|, where BA1_i_ and BA2_i_ denote the levels of a pair of BAs in individual *i*, and *n* is the total number of individuals.

The pairwise MAEs are presented graphically in Author response image 1. The average differences between two BAs range from 0.64 to 7.34 (SD). The highest MAE was observed between FI and DNAmGrimAge.

**Author response image 1. respfig1:** ****Mean absolute errors of BA pairs.

As MAE quantifies the absolute difference between two continuous variables, it would be an efficient indicator if all the BAs were assumed to measure the same underlying process, i.e., if the expected relationship between each pair of BAs is the identity line y=x. However, both prior biological knowledge and our results suggest that telomere length, DNA methylation age estimators, physiological age, and functional BAs are likely to measure different aspects of the aging process. Thus, assuming all the BAs are growing in the same manner would be a rather unrealistic assumption and makes MAE a less informative measure to indicate the difference between the two BA measures of the different types

Considering that we have already reported correlation coefficients to provide the information of how different BAs agree with each other, we will not include the MAE results in the manuscript and only present the figure in the response letter for the reviewer’s reference.

How many persons have different longitudinal time points with all the nine? If such patients exist, is it possible to give individual correlations of different BAs with age?

a) In total, 288 individuals had at least one complete measurement occasion, of which 105, 87, 60, 27, and 9 individuals had 1, 2, 3, 4, and 5 times of complete measurements.

Author response table 1: Number of individuals by number of complete measurements

Individuals withNumber of individualsNumber of complete measurements1 complete measurement1051052 complete measurements871743 complete measurements601804 complete measurements271085 complete measurements945any complete measurements288612

b) For the correlation analysis, we included all the 612 complete measurements and estimated the repeated measures correlation (rmcorr) coefficients, which accounts for non-independence among observations of the same individual.

Therefore, the reported correlation coefficients have already taken repeated measurements into consideration and are essentially measures of overall intra-individual correlation. We have expanded the methods description of the repeated measurement correlation in the manuscript to clarify this point:

“Using all complete measurements, we estimated repeated measures correlation (rmcorr, 22) between age and all nine BAs, which accounts for non-independence among observations and captures the common intra-individual association between age and each of the BAs in turn.”

Is it possible to conclude that longer telomeres or shorter DNAmAge are practically not associated with physiological indicators of functioning (clinical end-points)? Could the authors emphasise this aspect of the results in the Discussion?

It is indeed true that our results suggest molecular BAs are practically not associated with indicators of functioning (i.e., cognition, FAI, and FI) when we controlled for the age effect (Figure 2C). It can be interpreted that among the individuals with the same CA, molecular BAs and functional BAs are only weakly associated, and largely reflect different aspects of the aging process.

We have added the following sentences to emphasize this point in the Discussion section. Further interpretation regarding the characteristics of functional BAs in contrast to molecular BAs is detailed in the Discussion section (subsection “BA-mortality associations”, last paragraph).

“In other words, among individuals with the same CA, molecular BAs and functional BAs are only weakly associated. We interpret this as an indication that these categories largely reflect different aspects of the aging process.”

Reviewer #3:In this paper the authors assessed longitudinal measurements of multiple biological age in a population-based cohort from Sweden with long term follow-up. The dynamic changes of 4 types of 9 BAs over 20 years, mutual correlations and individual/joint associations with mortality of the BAs were explored. The study stands out in terms of longitudinal design, repeated measures of multiple types of BAs, and adequate statistical analysis. The results further confirmed the validity of multiple BAs as aging indicators and their potential complementary roles in aging process, and highlighted the variations with respect to gender. The following comments are mainly to improve the clarity.We thank the reviewer for this positive feedback and the following helpful comments to improve our work.1) Longitudinal changes of BAs over CA were investigated, whereas longitudinal changes of BAs in relation to mortality were not explored.

We agree with the reviewer that investigating the association between BA changes and mortality risk is an interesting and important topic. However, the current data is underpowered to test such effects as the number of individual with at least two measurements are quite limited, especially when it comes to telomere length and DNAmAge estimators (Table 2). Investigating the longitudinal change of functional BAs (cognitive function, FAI, and FI) in relation to health-related outcomes could be less affected by the power issue and there are, in fact, some on-going works from our research group devoted to this topic. Therefore, we have decided to not touch upon the topic in the present analysis.

Author response table 2: Number of individuals by number of BA measurements

Individuals with different number of measurementDNAmAgesTelomere lengthHorvathHannumPheno AgeGrim AgePhysiological ageCognitive functionFAIFI1196949494941241571271182137898989891261221591543140929292921421678895410680808080114917663557323232328111560546747477917746469658533983116914

2) Follow-up time for cognitive function and physiological age (and even FAI and FI) were obviously longer than follow-up time of other BAs, it therefore may not be appropriate to compare the strength of different BAs with mortality given that only the first BA assessment was included in the survival analysis. What are the results if truncating follow-up time for different BAs comparable?

We agree with the reviewer that different follow-up times might interfere with the comparability of multiple BAs in relation to mortality risk. In one of the sensitivity analyses, we replicated the individual BA-mortality association among the participants with the complete measurements, in which all baseline BA measures were investigated with the mortality risk during the same follow-up time, a median of 16.5 years. The results were present in Supplementary Table 10 in Supplementary file 1 and didn’t show major discrepancies compared to those in Table 4.

3) What are the health status of participants at baseline (first BA assessment), which may affect mortality risk and is suggested to be controlled in regression models.

We thank the reviewer for suggesting us to consider potential confounding effect due to baseline health status. We analyzed the self-reported data on prevalent diseases and found 4.0%, 1.7%, 5.5%, and 3.9% of the participants have reported previous diagnoses of heart failure, stroke, diabetes, and cancer, respectively, at their first available measurement occasion.

As a consequence, we have included Supplementary Table 11 in Supplementary file 1, and added the following sentence to the Results section.

“We also adjusted for the presence of the previous diseases, including heart failure, stroke, diabetes, and cancer in the survival model and the observed results were largely unchanged.”

4) Categories of BAs are suggested to be described explicitly in the Introduction.

We thank the reviewer for pointing this out and have now explicitly described the BA categories at the end of the first Introduction paragraph.

“One of the potential BAs, telomere length, is the length of a nucleotide sequence at the chromosomal ends, and represents the capacity for cell division (Mather et al., 2011). Epigenetic clocks comprise aging-related DNA methylation information from various genomic loci (i.e., CpG dinucleotides) (Hannum et al., 2013; Horvath, 2013). Another composite score derived from clinical measurements and blood biomarkers can be viewed as a physiological age measure (Finkel et al., 2019; Levine, 2013; Reynolds et al., 2005; Searle et al., 2008). Furthermore, organismal functioning status could reflect the biological aging process from various aspects, such as cognitive performance, physical functioning, and overall frailty (Finkel et al., 2019; Levine, 2013; Reynolds et al., 2005; Searle et al., 2008).”